# Global, Regional, and National Burden of Protein–Energy Malnutrition: A Systematic Analysis for the Global Burden of Disease Study

**DOI:** 10.3390/nu14132592

**Published:** 2022-06-22

**Authors:** Xu Zhang, Lu Zhang, Yuanchun Pu, Min Sun, Yan Zhao, Dan Zhang, Xin Wang, Yarui Li, Dan Guo, Shuixiang He

**Affiliations:** 1Department of Gastroenterology, The First Affiliated Hospital of Xi’an Jiaotong University, Xi’an 710061, China; zx347885709@stu.xjtu.edu.cn (X.Z.); sm980721@stu.xjtu.edu.cn (M.S.); yanzhao211@xjtufh.edu.cn (Y.Z.); zhangdan_h@xjtu.edu.cn (D.Z.); wangxin3@xjtufh.edu.cn (X.W.); liyarui0529@xjtufh.edu.cn (Y.L.); guodana@xjtufh.edu.cn (D.G.); 2Department of Urology, The First Affiliated Hospital of Xi’an Jiaotong University, Xi’an 710061, China; medicinezl1996@stu.xjtu.edu.cn (L.Z.); m13981250815@stu.xjtu.edu.cn (Y.P.)

**Keywords:** protein–energy malnutrition, global burden of disease, prevalence, death, disability-adjusted life years

## Abstract

Background: Statistical data on the prevalence, mortality, and disability-adjusted life years (DALYs) of protein–energy malnutrition are valuable for health resource planning and policy-making. We aimed to estimate protein–energy malnutrition burdens worldwide according to gender, age, and sociodemographic index (SDI) between 1990 and 2019. Methods: Detailed data on protein–energy malnutrition from 1990 to 2019 was extracted from the Global Burden of Disease (GBD) database. The global prevalence, deaths, and DALYs attributable to protein–energy malnutrition and the corresponding age-standardized rates (ASRs) were analyzed. Results: In 2019, the global prevalence of protein–energy malnutrition increased to 14,767,275 cases. The age-standardized prevalence rate (ASPR) showed an increasing trend between 1990 and 2019, while the age-standardized deaths rate (ASDR) and age-standardized DALYs rate presented a significantly decreasing trend in the same period. Meanwhile, there was a clearly ASPR, ASDR, and age-standardized DALYs rate downtrend of the prediction curve when the SDI went up. Conclusions: PEM still has a relatively serious disease burden in the world, especially in children and the elderly. At the same time, this phenomenon will be more obvious due to the aging of the world’s population. Effective prevention measures should be strengthened to continuously improve public health conditions.

## 1. Introduction

Malnutrition was defined as “a subacute or chronic state of nutrition, in which a combination of varying degrees of under- or overnutrition and inflammatory activity has led to changes in body composition and diminished function” [1,2]. Protein–Energy Malnutrition (PEM) is a series of diseases due to the malnutrition of all macronutrients, including marasmus, intermediate states of kwashiorkor-marasmus, and kwashiorkor. PEM is a common nutritional problem worldwide and can be seen in both developed and developing countries. The prevalence of the PEM in the older communities varied by region, from 0.8% to 24.6%, and it is also affected by the gender sampling frame and rurality [3]. The number of cases of children with PEM is declining globally but also varies by region; for example, it continues to decline in Asia but is increasing in Africa [4]. In addition, PEM poses a threat to public health, especially in children and the elderly, by impairing the immune response, which can lead to death [3,5]. Although the global incidence rate of protein–energy malnutrition has decreased in recent years [4], with the progress of medicine and health and the development of the food industry, it still causes an unavoidable health burden for all age groups. It is a top priority for us to make a huge contribution to solving such a public burden.

Some previous studies provided disease data for a specific population, yet there has been no research specifically addressing the global burden of protein–energy malnutrition and its changes. The purpose of this study was to assess the risk of protein–energy malnutrition based on global burden data and to describe the geographical location, age, gender, and SDI (socio-demographic index) of protein–energy malnutrition in 204 countries and territories from 1990 to 2019. We calculated ASPR (age-standardized prevalence rate), DALY (disability-adjusted life year), EAPC (estimated annual percentage change), and other indicators which were already used in different fields, such as cardiovascular diseases and pulmonary diseases, to clarify the different causes and trends of protein–energy malnutrition [6,7,8]. This study examines the various relationships of protein malnutrition in different regions and countries so that decision-makers can reasonably allocate social resources and release the social health burden of protein capacity malnutrition.

## 2. Materials and Methods

### 2.1. Data Source

The latest release of the Global Burden of Disease, Injuries, and Risk Factors Study (GBD) 2019 results was applied. The data was obtained on the official GBD website (http://ghdx.healthdata.org/gbd-2019 (accessed on 20 January 2022)) according to operation guidelines, without any inclusion or exclusion criteria. Data sources for the burden of protein–energy malnutrition were extracted with the Global Health Data Exchange (GHDx) query tool (http://ghdx.healthdata.org/gbd-results-tool (accessed on 20 January 2022)). Protein–energy malnutrition includes moderate and severe acute malnutrition, commonly referred to as “wasting,” and was defined in terms of weight-for-height Z-scores (WHZ) on the WHO 2006 growth standard for children. We quantified non-fatal PEM burden in four mutually exclusive and collectively exhaustive categories, reflecting distinct gradations of disability that can occur: moderate wasting without oedema (WHZ < −2 SD to < −3 SD), moderate wasting with oedema (WHZ < −2 SD to < −3 SD), severe wasting without oedema (WHZ < −3 SD), and severe wasting with oedema (WHZ < −3 SD). For PEM, ICD 10 codes are E40–E46.9 and E64.0, and ICD 9 codes are 260–263.9 [9]. We acquired data on the prevalence, mortality, and DALYs of protein–energy malnutrition and the respective age-standardized rate (ASR) of protein–energy malnutrition from 1990 to 2019. The 204 countries/territories were then categorized into five categories on the basis of the sociodemographic index (SDI) in 2019: high, high-middle, middle, low-middle, and low SDI. Besides, the human development index (HDI) values were derived from the World Bank.

The GBD estimation process is based on identifying multiple relevant data sources, including censuses, household surveys, civil registration and vital statistics, disease registries, health service use, and other sources. Each of these types of data is identified from a systematic review of published studies, searches of government and international organization websites, primary data sources such as the Demographic and Health Surveys, and contributions of datasets by GBD collaborators. First, individual-level and tabulated child anthropometry data from health surveys, literature, and national reports were used and centralized to inform the prevalence of weight-for-height Z-scores (WHZ) decrement in each category corresponding to our case definitions. Second, to inform the proportion of children under 5 years who have signs of organ failure manifested as oedema (i.e., kwashiorkor), a compiled dataset of surveys was conducted using Standardized Monitoring and Assessment of Relief and Transitions (SMART) methods. All data were extracted with the most detailed standard demographic identifiers available, including age, sex, country, year, and subnational location, if available [9].

### 2.2. Statistical Analysis

Estimated annual percentage change (EAPCs) and average annual percentage change (AAPCs) of age-standardized rates (ASRs) were calculated. We assumed that the natural logarithm of ASR was linear along with time; hence, Y=α+βX+ε (X represents the calendar year, Y represents $\ln\left(ASR\right)$), ε represents the error term, and β indicates the positive or negative trends of ASRs). The formula of the EAPC was $EAPC=100\times \left(exp\left(\beta \right)-1\right)$ and its 95% confidence intervals (CI) were calculated according to the linear model. When EAPC and the upper boundary of CI are negative, ASR represents a descending trend. In contrast, ASR is considered to be in an upward trend.

Average Annual Percent Change (AAPC) is a single number which represents the occurrence of disease in a population via applying the geometrically weighted averages for annual percent changes. The analysis was performed by the JointPoint Regression Program 4.9.0.1 (National Cancer Institute, Bethesda, MD, USA) provided by the United States National Cancer Institute Surveillance Research Program. To obtain the AAPCs, the software was applied to track trends in GBD data over time and then fit an underlying model possible to the data via connecting different line segments on a logarithmic scale. The segments are called “JointPoints”, and each is tested for significance by a Monte Carlo permutation method. The analyses were performed using R statistical software version 4.1.2 (https://www.r-project.org (accessed on 25 February 2022)). *p* < 0.05 was regarded as statistically significant.

## 3. Results

### 3.1. Global Burden of Protein–Energy Malnutrition

In 2019, there were 14,767,275 (95% CI = 130,405,924–167,471,360) prevalence cases, and the age-standardized prevalence rate (ASPR) was 2006.4 (95% CI = 1786–2261.3) per 100,000 population. It is noteworthy that ASPR EAPC showed an increasing trend of, on average, 0.19% (95% CI = 0.08–0.31%) per year (Table 1).

There were 212,242 (95% CI = 185,403–246,217) death cases and the ASDR (age-standardized deaths rate) was 3 (95% CI = 2.6–3.5) per 100,000 population. Conversely, the EAPCs of the ASDR was −5.15% (95% CI = −5.5%–4.8%), which shows a decrease trend (Appendix A).

At the same time, PEM led 15,256,524 (95% CI = 12,565,114–18,327,803) DALYs with an age-standardized rate of DALYs 218.3 (95% CI = 179.5–262.8) per 100,000 population. The age-standardized DALYs rate shows a significantly decreasing trend in the same period, −5.03% (95% CI = −5.27–4.79%) (Appendix A).

### 3.2. Regional Burden of Protein–Energy Malnutrition

For different regions, in 2019, the highest ASPRs (per 100,000) appeared in South Asia (3316.7 (95% CI = 2961.3–3752.9)), Southeast Asia (2563.6 (95% CI = 2297.1–2876.1)), Oceania (1780.8 (95% CI = 1647.1–1922.7)), East Asia (1731.3 (95% CI = 1425.6–2098)), and Western Sub-Saharan Africa (1690.4 (95% CI = 1572.9–1821.5)), with the lowest ASPRs in Australasia (522.9 (95% CI = 444.9–618.2)), Andean Latin America (573.8 (95% CI = 513.2–638.7)), and Tropical Latin America (694.2 (95% CI = 613.7–790)). The biggest increase trend was in East Asia (1.05 (95% CI = 0.91–1.19)), and the biggest decrease trend was in Andean Latin America (−0.92 (−1.08–0.75) (Table 1)).

In the ASDR (per 100,000) part, the highest value was Eastern Sub-Saharan Africa (17.6 (95% CI = 14.6–20.7)), with the lowest values in Eastern Europe (0.1 (95% CI = 0.1–0.1)), Central Europe (0.2 (95% CI = 0.1–0.2)), and Central Asia (0.2 (95% CI = 0.2–0.2)). The highest trend appeared in Central Europe (1.06 (95% CI = 0.5–1.63)), with the lowest trend in East Asia (−8.82 (95% CI = −11.27–6.31)) (Appendix A).

The highest age-standardized rate of DALYs appeared in Eastern Sub-Saharan Africa (716.6 (95% CI = 585.7–886)), and the lowest were in Australasia (18.3 (95% CI = 12.6–25.8)) and High-income Asia Pacific (20.2 (95% CI = 14.5–27.6)). The EAPC had the highest score in Western Europe (0.44 (95% CI = 0.3–0.57)), which showed the biggest increase trend. Relatively, the biggest decrease trend appeared in East Asia (−8.99 (95% CI = −11.16–6.77)) (Appendix A).

For the AAPC of prevalence, East Asia was top, High−middle SDI second, with Eastern Sub−Saharan Africa at the bottom (Figure 1A). In the AAPC of deaths, most regions had a negative number, with Central Europe, High−income North America, Australasia, Western Europe, and South Asia possessing the most negative number simultaneously (Figure 1B). As for the AAPC of DALYs, Western Europe had the biggest value, and South Asia had the smallest (Figure 1C). The relative values of DALYs are YLLs and YLDs. Only High−income North America achieved a positive value in the AAPC of YLLs, and East Asia received the smallest negative number. However, East Asia got the biggest AAPC of YLDs. Eastern Sub−Saharan Africa was at the bottom in the AAPC of YLDs (Appendix A).

### 3.3. National Burden of Protein–Energy Malnutrition

At the national level, the Maldives had the highest number of ASPR in 2019, whereas Mongolia and Peru had the smallest number (Figure 2A). The Czech Republic had the biggest EAPC of prevalence, and Guatemala had the smallest (Figure 1B). Mali had the highest number of ASDRs (Appendix A). The Czech Republic had the biggest EAPC of ASDR, and the Democratic People’s Republic of Korea had the smallest (Appendix A). Eritrea had the highest age-standardized DALYs rate, while Singapore had the smallest number (Appendix A). The Czech Republic had the biggest EAPC of DALYs, and Cambodia had the smallest (Appendix A).

### 3.4. Age and Sex

For the prevalence rate, the prevalence basically increased with age, except children aged 1–4 had the highest prevalence (Figure 3A). For the death rate, the prevalence increased significantly with age, peaking at 95+, except in children aged 1–4 (Figure 3B). With the DALYs rate, the value of minors decreased with age, the value of adults increased significantly with age, and female children aged 1–4 years were significantly higher than male children (Figure 3C). The trend of the DALYs rate also appeared in the YLL rate and YLD rate (Appendix A).

### 3.5. The Socio-Demographic Index (SDI) and Human Development Index (HDI)

SDI refers to the socio-demographic index, and HDI refers to the human development index. Their values can be used to judge the degree of economic development of a country or region so as to compare the relationship and causes between protein malnutrition and the level of national economic development.

For the relationship between prevalence and SDI, there was clearly an ASPR downtrend of the prediction curve when the SDI went up (R = −0.59, *p* < 2.2 × 10^−16^. At the regional level, the ASPRs of Global, South Asia, Southeast Asia, and Western Europe were higher than predicted, and the Southern Sub-Saharan Africa, Central Asia, High-income Asia Pacific, Southern Latin America, Oceania, Caribbean, Andean Latin America, High-income North America, Central Latin America, Tropical Latin America, Australasia, and Central Sub-Saharan Africa ASPRs were lower than predicted (Figure 4). At the national level, it also showed a negative trend between prevalence and SDI (R = −0.33, *p* = 2 × 10^−6^) (Appendix A). The EAPC of prevalence was negatively correlated with SDI (*p* = −0.53 (95% Cl: = −0.63–0.41), *p* < 0.001) (Figure 5A).

For the relationship between deaths and SDI, there was clearly an ASDR downtrend of the prediction curve when the SDI went up (R = −0.73, *p* < 2.2 × 10^−16^). At the regional level, the ASDRs of Global, Southern Sub-Saharan Africa, Eastern Sub-Saharan Africa, Andean Latin America, Central Latin America, Tropical Latin America, Central Sub-Saharan Africa, and Southeast Asia were higher than predicted, and Central Asia, South Asia, Oceania, North Africa, and the Middle East were lower than predicted (Appendix A). At the national level, it also showed a negative trend between deaths and SDI (R = −0.69, *p* < 2.2 × 10^−16^) (Appendix A). The EAPC of deaths was negatively correlated with SDI (*p* = −0.42 (95% Cl: −0.53–0.29), *p* < 0.001) (Figure 5B).

For the relationship between DALYs and SDI, there was clearly an age-standardized DALYs rate downtrend of the prediction curve when the SDI went up (R = −0.73, *p* < 2.2 × 10^−16^). At the regional level, the age-standardized DALYs rates of Global, Southern Sub-Saharan Africa, Eastern Europe, the Caribbean, Central Sub-Saharan Africa, and Western Europe were higher than predicted, and Central Asia, Oceania, North Africa, and the Middle East were lower than predicted (Appendix A). At the national level, it also showed a negative trend between deaths and SDI (R = −0.66, *p* < 2.2 × 10^−16^) (Appendix A). The EAPC of DALYs was negatively correlated with SDI (*p* = −0.64 (95% Cl: −0.72–0.54), *p* < 0.001) (Figure 5C).

For the relationship between EAPC and HDI, EAPC of prevalence was negatively correlated with HDI (*p* = −0.53 (95% Cl: −0.63–0.41), *p* < 0.001) (Appendix A). The EAPC of deaths was negatively correlated with HDI (*p* = −0.42 (95% Cl: −0.54–0.29), *p* < 0.001) (Appendix A). The EAPC of DALYs was negatively correlated with HDI (*p* = −0.64 (95% Cl: −0.72–0.55), *p* < 0.001) (Appendix A).

We divided the SDI into five levels: High SDI, High-middle SDI, Middle SDI, Low-middle SDI, and Low SDI. The number of prevalence cases in High SDI and High-middle SDI regions was significantly lower than that in Middle SDI, Low-middle SDI, and Low SDI regions. The High SDI and High-middle SDI regions’ cases were dominated by people over the age of 15, while the Middle SDI, Low-middle SDI, and Low SDI regions’ cases were dominated by people under the age of 15. Meanwhile, the Low−middle SDI region had the highest number of cases in children under 5 years of age (Figure 6). As for the death cases, the global death cases were mainly children under 5 years old. The Low-middle SDI and Low SDI regions had the largest number of overall cases, but they are decreasing year by year. In terms of age composition, the High SDI had always been dominated by the 75+ age group, and the High-middle SDI and Middle SDI areas gradually transitioned from children under 5 years old to 75+ over time, and Low-middle SDI and Low SDI areas had always been mainly for children under 5 years old. In particular, between 1995 and 2002, the total number of cases in the Low-middle SDI region increased abnormally (Appendix A). For DALYs cases, the total number of cases in the world and regions decreased other than High SDI. Among them, Low-middle SDI and Low SDI had the largest total number of cases, and the High SDI region had the smallest total number of cases. In terms of age composition, the Middle SDI, Low−middle SDI, and Low SDI regions were dominated by the age group under 5 years old. The proportion of people under the age of 5 in the High−middle SDI and Middle SDI regions has been decreasing. The Low−middle SDI region also increased abnormally between 1995 and 2002 (Appendix A).

In the JointPoint figure, the prevalence of Global, High SDI, High-middle SDI, and Middle SDI increased from 1990 to 2011, and the Low-middle SDI and Low SDI decreased at the same time. The prevalence of all SDI groups went through a U-shaped curve from 2011 to 2019 (Figure 7; Appendix A). As for the JointPoint of deaths, Global, High-middle SDI, Middle SDI, and Low SDI decreased from 1990 to 2019, and the High SDI and Low-middle SDI also decreased from 2000 to 2019 (Figure 7; Appendix A). The JointPoint of all SDI groups except the High SDI decreased over time (Figure 7; Appendix A).

## 4. Discussion

This study is the first experiment to use the GBD2019 database to investigate the global burden of PEM. We comprehensively assessed the prevalence, deaths, DALYs, YLLs, YLDs, and corresponding ASRs of PEM and compared data across countries, regions, age, sex, and SDI. Globally, both the ASDR and age-standardized rate of DALY decreased from 1990 to 2019, but ASPR increased slightly. There is a relationship between economic level and malnutrition [10,11]. The improvement of the economic level and medical science can cause the improvement of the expected life span, which could increase in ASPR of PEM. At the same time, because the PEM diagnostic standards continuously improved between 1990 and 2019, the screening of PEM is more comprehensive, which may lead to the continuous increase in PEM’s ASPR, and also lead to abnormal turning points in the JointPoint diagrams [2,12,13]. The decline in ASDR and ASR DALYS can also be attributed to the improvement of medical technology.

The burden of PEM has an obvious relationship with the socioeconomic level. Socioeconomic indicators HDI and SDI were negatively correlated with PEM burdens, which means that higher socioeconomic levels have lower PEM burdens. High HDI represents a higher level of prevention and treatment, and higher SDI represents a higher social security capability. These are all favorable factors to reduce the burden of PEM.

The age distribution of patients with PEM has a good concentration. We found that PEM patients are mainly concentrated in the children group and older group. For the children group, PEM is the most serious malnutrition in children. Child patients generally lack the amino acids required for growth and development. The lack of amino acids will affect the normal growth of cells and collectives. For example, the lack of amino acids will cause the main growth regulation of cells, Mechanical Target of Rapamycin Complex 1 (MTORC1), to synthesize and inhibit the growth of cells and the body, which leads to slow development and immune function defects [14,15,16]. PEM causes 56% of children’s deaths in developing countries [17]. The main risk factors for children’s PEM are environmental, poverty, gender, food culture, and immunization [16,18,19,20], which will be reflected in the burden level of different regions. However, the high SDI place can guarantee a better nutritional level for children. Compared with the condition of the children group, the elderly group is different. PEM can be seen in about 50% of the elderly in hospitalization. With the increase in age, the elderly groups have gradually experienced a loss of appetite, insufficient diet, and weakened intestinal absorption capacity, which can easily lead to PEM. The elderly with PEM are prone to decreased concentration of ornithine, histidine, glutamic acid, and glutamine, which leads to the symptoms of decreased physical function and decreased immune function, which eventually leads to death [21,22]. Therefore, government and medical institutions need to pay special attention to the development of children and elderly PEM, actively test relevant indicators, and formulate positive strategies to prevent death.

As in this experiment, women had been found to be at a higher risk of PEM than men in several studies [23,24], but no specific studies have demonstrated whether the specific reasons are related to gender inequality, physiological differences, or life expectancy, etc. [3]. This also suggests the direction of our future research in order to provide women with more preventive and therapeutic measures.

At the regional level, ASPR, ASDR, age-standardized DALYs rate, EAPC, and AAPC vary from place to place. South Asia, Southeast Asia, and East Asia occupy the top positions in the ASPR rankings, while East Asia also has the highest EAPC of ASPR, the lowest EAPC of ASDR, the lowest EAPC of DALYS, and the highest AAPC of prevalence. The rapid rise of ASPR in East Asia may be related to the general rapid aging of countries in East Asia [25,26]. Moreover, the elderly are more likely to obtain PEM. At the same time, South Asia, Southeast Asia, and East Asia are generally developing countries, and the ability to provide formal medical services is not as good as that of developed countries [27]. However, in East Asian countries such as China, the number of PEM has decreased significantly in recent years due to rapid economic development, the substantial improvement in the quality of medical services, and major efforts in PEM prevention [28]. At the same time, China has the largest population base in East Asia, so it is the most likely reason for the decline in ASDR and DALYs [3]. Rapidly developing economies and incomplete healthcare systems also pose enormous challenges to PEM public health services in East Asia [26]. Although Eastern Sub-Saharan Africa had the lowest AAPC of prevalence, it had the highest ASDR and age-standardized DALYs rate, which is closely related to the lower total population, economic level, and medical level of this region [29,30]. At the same time, maize as the main energy source in Eastern Sub-Saharan Africa also leads to the apparent lack of tryptophan and lysine in the region [31]. This region needs certain dietary guidance policies to promote the reduction in the PEM burden.

This study has performed a comprehensive review of the PEM burden at the global, regional, and national levels, but there exist some limitations. Because of the complexity and breadth of the data, the PEM burden should be interpreted cautiously. Fewer data are available from countries and regions with lower SDI values, and the disease burden may be underestimated because of different levels of registration management. Moreover, though the data of the GBD study are considered of high quality, differences in data collecting, extracting, coding, and quality of data sources inevitably compromise the robustness and accuracy of GBD estimates. Obviously, the quality of data collection coding and data sources is qualitatively related to the economic and political level of countries and regions. In developed countries, especially developed countries with relatively complete medical systems, such as the United Kingdom, the United States, and France, the data obtained are relatively reliable and complete, while in some developing countries, such as most African and Central Asian countries, due to imperfections in the economy, war, or the imperfection of the medical system, statistics may be lost or wrong. At the same time, the true burden of the disease may be underestimated due to the difficulty of diagnosing PEM. Last but not least, the fluctuations in disease burden data may partly represent the detection bias associated with adjustments in screening protocols rather than real changes.

## 5. Conclusions

This study provides a reference for monitoring the burden and trends of PEM at the global, regional, and national levels, which is crucial to conducting interventions that might slow down the rising burden of PEM. Our findings found that PEM still has a relatively serious disease burden in the world, especially in lower SDI regions, children, and the elderly. Consequently, more attention should be paid to developing early prevention and treatment measures for PEM in lower SDI regions and countries, such as safeguarding the food supply, eliminating hunger, and improving overall nutritional status. At the same time, particular attention to children and elderly PEM is needed. Effective steps, such as enhancing nutrition-related health education, strengthening nutritional support, and early aggressive treatment, should be formulated to relieve the burden of PEM. Further studies are required to identify more useful public health interventions.

## Figures and Tables

**Figure 1 nutrients-14-02592-f001:**
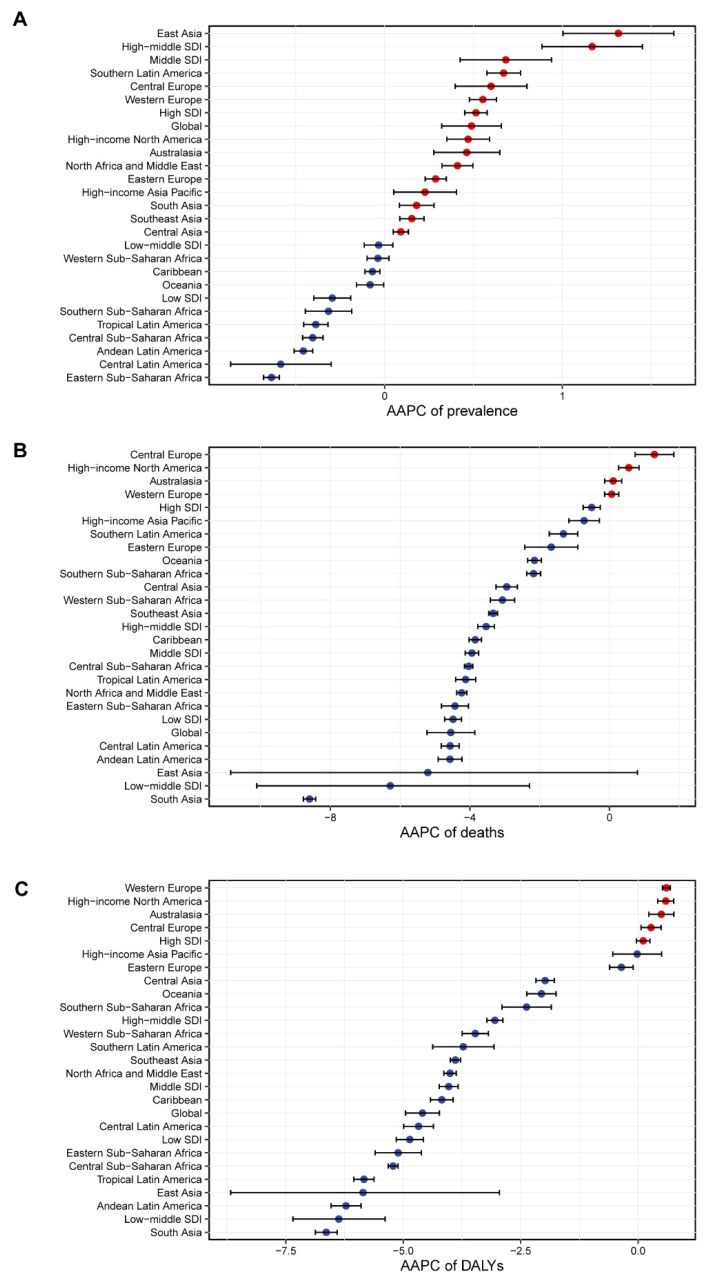
Average annual percentage change (AAPC) of the (**A**) age-standardized prevalence rate, (**B**) age-standardized death rate, and (**C**) age-standardized disability-adjusted life year rates for protein–energy malnutrition from 1990 to 2019. AAPC was obtained representing the average percent increase or decrease in PEM rates per year over each specified period of time to summarize and compare these trends over the entire time period. Red dot represents a AAPC value greater than zero, while blue dot represents a AAPC value less than zero.

**Figure 2 nutrients-14-02592-f002:**
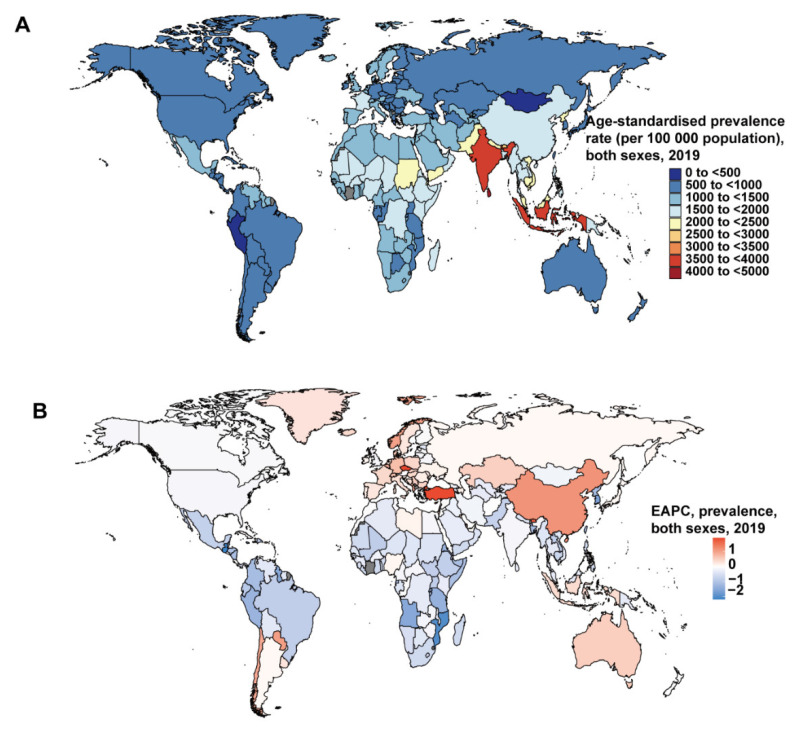
The global disease burden of protein–energy malnutrition for both genders in 204 countries and territories. (**A**)The age-standardized prevalence rate (ASPR) of protein–energy malnutrition in 2019. (**B**) The estimated annual percentage change (EAPC) in ASPR of protein–energy malnutrition from 1990 to 2019.

**Figure 3 nutrients-14-02592-f003:**
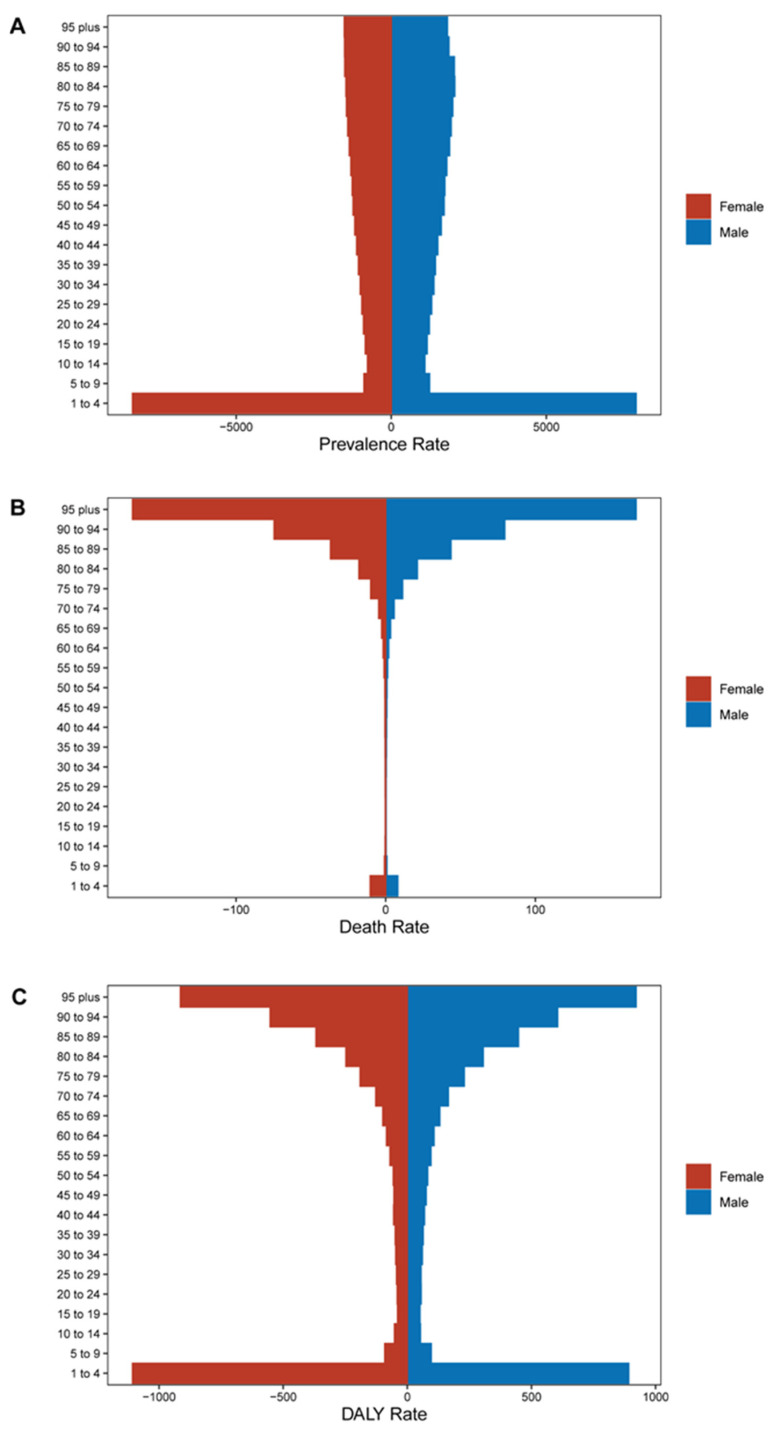
The prevalence, death, and disability-adjusted life year (DALY) rates of protein–energy malnutrition among gender and age. (**A**) Prevalence. (**B**) Death rate. (**C**) DALY rate.

**Figure 4 nutrients-14-02592-f004:**
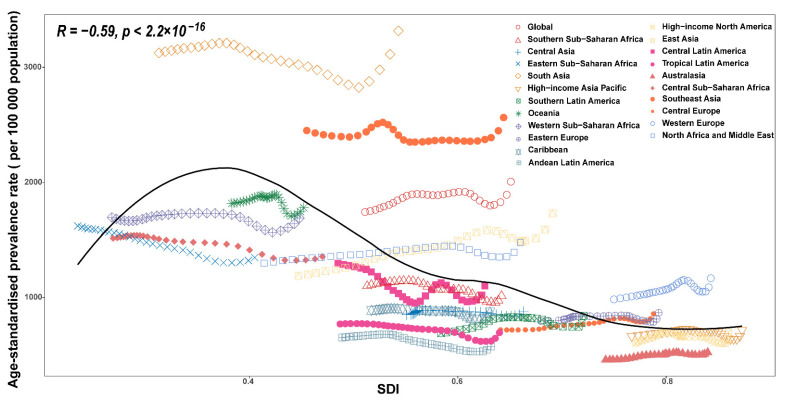
Age-standardized prevalence rates for protein–energy malnutrition for 21 GBD regions by socio-demographic index (SDI), 1990–2019. There was clearly an ASPR downtrend of the prediction curve when the SDI went up. Expected values based on the socio-demographic index and disease rates in all locations are shown as the black line.

**Figure 5 nutrients-14-02592-f005:**
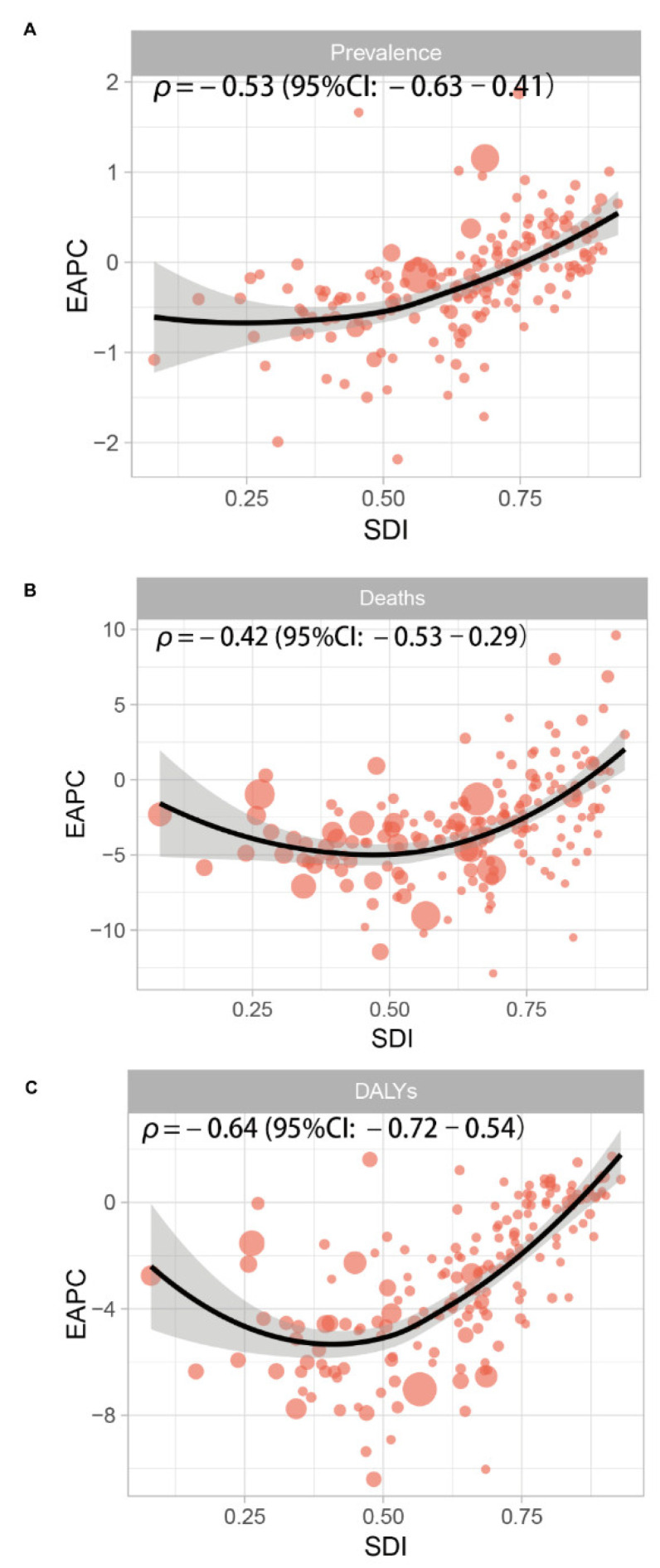
The association between the socio-demographic index (SDI) and estimated annual percentage change (EAPC) in age-standardized prevalence rate (ASPR), age-standardized death rate (ASDR), and age-standardized disability-adjusted life years of protein–energy malnutrition. SDI (2019) was negatively associated with EAPC in (**A**) ASPR (ρ = −0.53, *p* < 0.001), (**B**) APSR (ρ = −0.42, *p* < 0.001), and (**C**) age-standardized DALY (ρ = −0.64, *p* < 0.001).

**Figure 6 nutrients-14-02592-f006:**
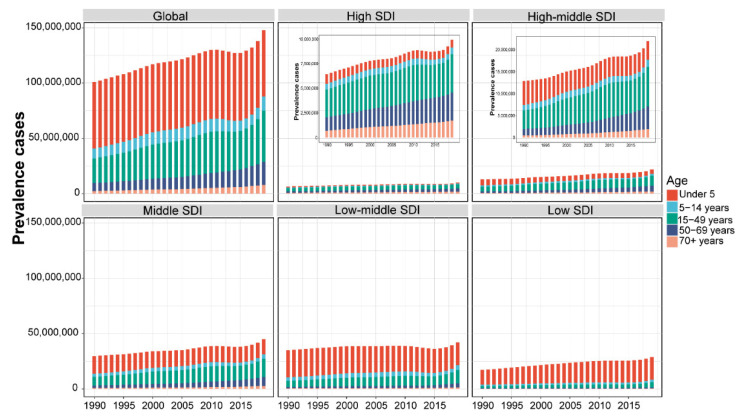
The proportion of the five age groups for protein–energy malnutrition prevalence cases between 1990 and 2019 globally, and in High, High-middle, Middle, Low-middle, and Low SDI quintiles. The populations were divided into five age groups: under 5, 5–14 years, 15–49 years, 50–69 years, and 70+ years. SDI values can be used to judge the degree of economic development of a country or region.

**Figure 7 nutrients-14-02592-f007:**
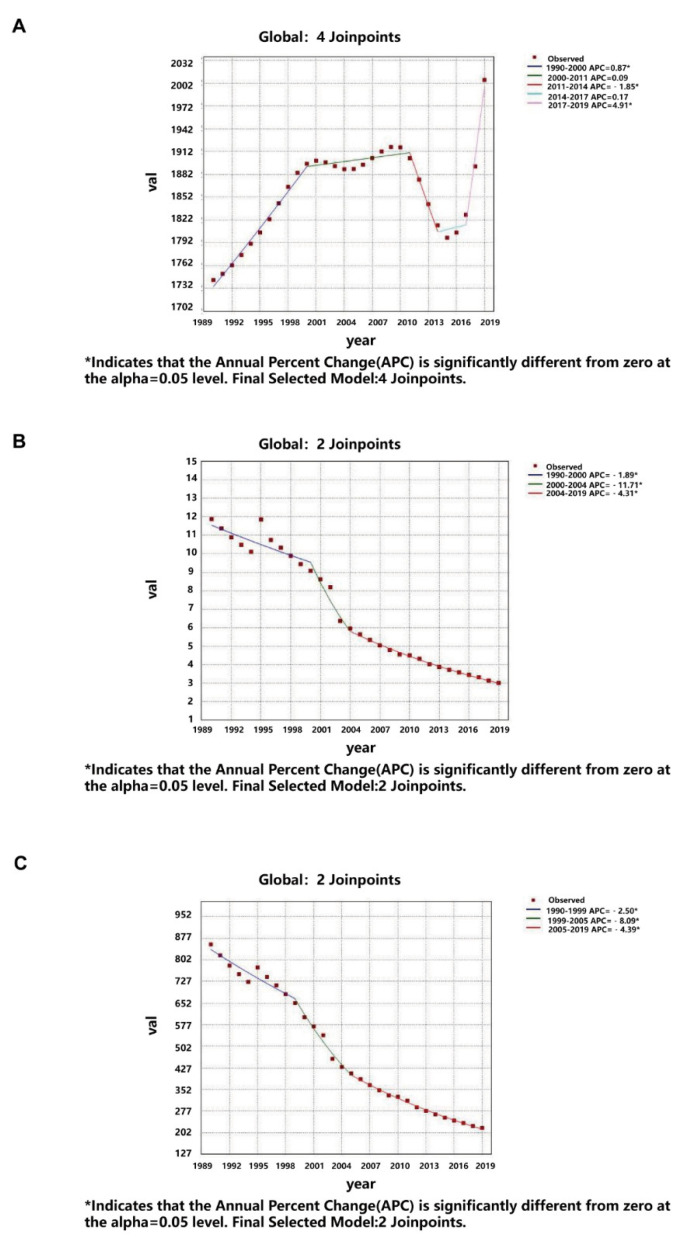
JointPoint regression analysis for protein–energy malnutrition from 1990 to 2019. (**A**) Age-standardized prevalence rate globally; (**B**) Age-standardized death rate globally; (**C**) Age-standardized disability-adjusted life years globally. The prevalence of protein–energy malnutrition increased from 1990 to 2011 globally, and went through a U-shaped curve from 2011 to 2019. The JointPoint of deaths and DALYs decreased between 2000 and 2019.

**Table 1 nutrients-14-02592-t001:** The prevalence cases and age-standardized prevalence of PEM in 1990 and 2019, and its temporal trends from 1990.

	1990		2019		1990–2019
	Prevalence Cases No. (95% UI)	ASR per 100,000 No. (95% UI)	Prevalence Cases No. (95% UI)	ASR per 100,000 No. (95% UI)	EAPC No. (95% CI)
Global	100,977,331.3 (92,783,224.2–110,877,370.1)	1743.1 (1587.1–1928.7)	147,672,757.9 (130,405,923.7–167,471,359.5)	2006.4 (1786–2261.3)	0.19 (0.08–0.31)
Social-demographic index					
Low SDI	17,156,823.7 (16,449,965.2–18,031,793.4)	2218.2 (2076.1–2387.2)	28,682,621.5 (26,973,402.2–30,794,848.6)	2030.2 (1872.4–2221.8)	−0.61 (−0.72–0.51)
Low-middle SDI	34,823,174.6 (32,775,266.5–37,362,791.2)	2445.7 (2262.2–2670.4)	42,123,907.2 (37,747,744.6–47,459,684)	2429.1 (2186.2–2723.2)	−0.45 (−0.58–0.32)
Middle SDI	29,621,821.1 (26,779,018.9–33,164,070.2)	1648.3 (1479.7–1848.9)	44,926,275.8 (38,798,928.9–52,033,671.7)	1996.2 (1748.7–2288.2)	0.42 (0.31–0.52)
High-middle SDI	12,900,075.5 (11,312,222.2–14,908,449.5)	1166 (1029.2–1337.5)	21,951,415 (18,393,911.9–26,054,900.4)	1638 (1401.6–1919.3)	0.98 (0.85–1.1)
High SDI	6,429,788.8 (5,347,783.1–7,673,702.3)	788.6 (661.2–944.7)	9,921,037.9 (8,203,301.8–11,768,169.2)	923.1 (765.9–1105.5)	0.2 (0.08–0.33)
Region					
Central Asia	745,903.7 (702,157.8–794,880.2)	849.7 (789.3–920)	825,914 (754,870–909,758.1)	874.5 (799.9–962.6)	−0.07 (−0.15–0)
Central Europe	760,858.2 (641,646.6–908,940.9)	718.7 (621.9–839.5)	826,696.3 (689,623.4–991,992.1)	858 (734.7–1012)	0.53 (0.45–0.61)
Eastern Europe	1,514,453.3 (1,325,633.1–1,750,709.8)	795.2 (708.8–899.2)	1,374,636.6 (1,171,915.8–1,626,208.8)	866.9 (764.9–997.1)	0.04 (−0.06–0.14)
Australasia	86,161.7 (71,127.9–102,835.3)	457.8 (384.1–541.3)	147,972.6 (123,728.6–176,502.4)	522.9 (444.9–618.2)	0.41 (0.32–0.51)
High-income Asia Pacific	1,021,487.4 (853,742.1–1,224,002.9)	664.5 (571.2–779.4)	1,218,063 (1,003,598.5–1,453,380.1)	716.6 (607.3–850)	−0.15 (−0.31–0.01)
High-income North America	1,747,931 (1,392,721.8–2,150,830.2)	603.5 (481.9–748.9)	2,819,995.5 (2,271,045.4–3,422,009.3)	699.6 (558.7–863.1)	−0.09 (−0.27–0.1)
Southern Latin America	338,303.3 (293,896.6–390,650.5)	688.6 (596.8–795.6)	564,920.9 (477,318.1–666,322.6)	840.5 (715.9–985.5)	0.29 (0.06–0.52)
Western Europe	3,851,175.4 (3,163,177.4–4,659,506)	983.4 (811.4–1201)	5,681,745.9 (4,686,331.5–6,704,396.3)	1167.2 (959.1–1407.5)	0.39 (0.26–0.52)
Andean Latin America	240,068.3 (218,594.8–261,646.9)	652.1 (583.5–720)	355,853.7 (318,216.1–396,580.6)	573.8 (513.2–638.7)	−0.92 (−1.08–0.75)
Caribbean	325,151.1 (298,094.6–356,433.9)	887.2 (805.6–976.6)	390,393.7 (347,659.3–436,540.6)	873.3 (785.8–970.6)	−0.32 (−0.41–0.23)
Central Latin America	2,025,781.6 (1,806,272–2,281,193.9)	1296.3 (1134.6–1475)	2,681,644.3 (2,293,249.8–3,139,277.2)	1102.6 (948.2–1282.6)	−0.84 (−1.14–0.55)
Tropical Latin America	1,159,104.5 (1,044,944.5–1,292,708.6)	769.7 (687.3–864.7)	1,415,034.7 (1,225,954.6–1,634,539.4)	694.2 (613.7–790)	−0.76 (−0.9–0.63)
North Africa and Middle East	5,892,716.2 (5,553,048.4–6,340,665.1)	1298 (1199.3–1424.7)	8,814,709.6 (7,949,141.2–9,906,526.1)	1477.4 (1335.7–1649.6)	0.23 (0.12–0.34)
South Asia	43,696,425.7 (40,926,199.5–47,183,646.5)	3125 (2874.5–3435.9)	57,490,327.3 (50,908,475.1–65,795,679.3)	3316.7 (2961.3–3752.9)	−0.27 (−0.4–0.13)
East Asia	14,029,975.9 (11,926,576.9–16,629,041.4)	1190 (1013–1403.5)	26,118,724.4 (21,263,244.6–31,657,526)	1731.3 (1425.6–2098)	1.05 (0.91–1.19)
Oceania	147,114.8 (139,460.9–155,676)	1817.8 (1690.5–1947.6)	283,672.2 (266,582.1–302,257.4)	1780.8 (1647.1–1922.7)	−0.18 (−0.3–0.06)
Southeast Asia	12,093,977.1 (11,147,165.8–13,222,689.6)	2450.7 (2227.3–2707.1)	16,142,385.4 (14,315,450.9–18,265,959.5)	2563.6 (2297.1–2876.1)	−0.07 (−0.17–0.03)
Central Sub-Saharan Africa	1,332,271.9 (1,282,265.5–1,387,876.2)	1514.5 (1434.8–1600.2)	2,394,812.8 (2,282,754.9–2,514,402.1)	1355.2 (1274.6–1440.8)	−0.54 (−0.65–0.44)
Eastern Sub-Saharan Africa	4,387,305.3 (4,215,311.8–4,574,173)	1621.8 (1513.8–1740)	7,009,047.6 (6,721,523.8–7,341,028.7)	1347.4 (1266–1438.5)	−0.84 (−0.91–0.76)
Southern Sub-Saharan Africa	647,947.4 (603,654.8–699,495)	1098.7 (1006.1–1202.6)	789,790.3 (724,192.5–864,048)	1012.4 (925.9–1107.4)	−0.59 (−0.71–0.47)
Western Sub-Saharan Africa	4,933,217.4 (4,750,330.9–5,167,049)	1695.9 (1592.7–1819)	10,326,417 (9,839,148.9–10,911,512.6)	1690.4 (1572.9–1821.5)	−0.12 (−0.24–0.01)

## Data Availability

Data are available in a public, open access repository. All of the data are publically available. Data are available on request.

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
