# Peer review of "Global, Regional, and National Burden of Protein–Energy Malnutrition: A Systematic Analysis for the Global Burden of Disease Study"

_nutrients, 2022, doi:10.3390/nu14132592_

Round 1

Reviewer 1 Report

Please improve and clarify the following issues:

1)      In my opinion, name ‘The Czech Republic’ is better in official documents than ‘Czechia’.

2)      I do not understand the order of the figures, e.g. why is 5 and 6 first, and then 1, 2? Some of the figures have good resolution and are clear (e.g. Figures 2 and 5), others need to be improved to make them legible (e.g. Figures 1, 3, 6, and 7). Please, organize the appearance, quality and order of the figures.

3)      The authors have done a great job of reviewing the data, but I have the feeling that the numbers are a bit under-estimated. Therefore, I would like to ask the authors what are the limitations and weaknesses of this work? Is there any alternative to the presented statistical research?

4)      I believe that the conclusions should be written in more detail, emphasizing their role in improving the problem of malnutrition in the world.

Reviewer 2 Report

This is a data-extraction exercise that used the GBD2019 database to investigate the global burden of PEM. The authors comprehensively assessed the prevalence, deaths, disability-adjusted life years, years-of-life lost, years lived with disability, and corresponding age-standardized rates of PEM, and compared data across countries, regions, age, sex, and socio demographic index. I am not a biostatistitian. Few readers of this journal are statistical experts. This manuscript is nevertheless submitted to this journal, rather than a narrower epidemiological journal. 

A non-expert, I cannot vouch for or attest to the statistical validity of the material in the manuscript, but the procedure used seems completely straightforward, with some provisos indicated below.  The resulting analysis seems sound, and useful as a reference source. No outstanding novel findings are reported, nor would they be expected. 

Comments: 

1. There is a typo in abtract: SDI: Yocio-demographic should be Socio-demographic.

2. Please explain to readers how the information presented here differs from the information already provided in refs 3 and 4. Is it a different class or kind of information? What is novel or different from the information in refs 3 and 4. Do the conclusions differ or agree? Expand on this if indicated.  

3. Regarding the identification of the data, and the Global Health Data Exchange (GHDx) query tool: please specifically state how this tool defines PEM, as it was used here to identify the data used in this analysis. 

4. Please also explain the PROCEDURE by which the GBD obtains the PEM data that were extracted for this analysis. For example, are they determined from formal population surveys of ambulatory people, not hospitalized? Were hospitalized patients included in this data set? Were people in active famine zones included or excluded? 

5. Most of the figures in the manuscript are far too miniaturized to read. 

6. In the interests of making this material more inviting and intelligible to average readers of this journal, vanishingly few of whom are biostatisticians, the figure legends should be greatly improved to explain in plain terms what each figure indicates. Identify the axes routinely whenever there is the possibility of doubt about what is being indicated. Make the figures intelligible. Each figure ought to be explained sufficiently well, in its legend, that an unsophisticated, average reader of this journal will promptly understand what it is displaying.  

7. Only include figures that are central to the key messages, and relegate unimportant ones to supplementary files. 
